# Detection of Antibiotic Resistance in Feline-Origin ESBL *Escherichia coli* from Different Areas of China and the Resistance Elimination of Garlic Oil to Cefquinome on ESBL *E. coli*

**DOI:** 10.3390/ijms24119627

**Published:** 2023-06-01

**Authors:** Yin-Chao Tong, Peng-Cheng Li, Yang Yang, Qing-Yi Lin, Jin-Tong Liu, Yi-Nuo Gao, Yi-Ning Zhang, Shuo Jin, Su-Zhu Qing, Fu-Shan Xing, Yun-Peng Fan, Ying-Qiu Liu, Wei-Ling Wang, Wei-Min Zhang, Wu-Ren Ma

**Affiliations:** 1College of Veterinary Medicine, Northwest A&F University, Yangling 712100, China; 2College of Animal Science and Technology, Northwest A&F University, Yangling 712100, China; 3Institute of Traditional Chinese Veterinary Medicine, Northwest A&F University, Yangling 712100, China; 4College of Chemistry & Pharmacy, Northwest A&F University, Yangling 712100, China; 5Xi’an Veterinary Teaching Hospital, Northwest A&F University, Xi’an 710065, China

**Keywords:** feline origin, ESBL *E. coli*, garlic oil, antibiotics resistance elimination, cefquinome

## Abstract

The development of drug-resistance in the opportunistic pathogen *Escherichia coli* has become a global public health concern. Due to the share of similar flora between pets and their owners, the detection of pet-origin antibiotic-resistant *E. coli* is necessary. This study aimed to detect the prevalence of feline-origin ESBL *E. coli* in China and to explore the resistance elimination effect of garlic oil to cefquinome on ESBL *E. coli*. Cat fecal samples were collected from animal hospitals. The *E. coli* isolates were separated and purified by indicator media and polymerase chain reaction (PCR). ESBL genes were detected by PCR and Sanger sequencing. The MICs were determined. The synergistic effect of garlic oil and cefquinome against ESBL *E. coli* was investigated by checkerboard assays, time-kill and growth curves, drug-resistance curves, PI and NPN staining, and a scanning electronic microscope. A total of 80 *E. coli* strains were isolated from 101 fecal samples. The rate of ESBL *E. coli* was 52.5% (42/80). The prevailing ESBL genotypes in China were *CTX-M-1*, *CTX-M-14*, and *TEM-116*. In ESBL *E. coli*, garlic oil increased the susceptibility to cefquinome with FICIs from 0.2 to 0.7 and enhanced the killing effect of cefquinome with membrane destruction. Resistance to cefquinome decreased with treatment of garlic oil after 15 generations. Our study indicates that ESBL *E. coli* has been detected in cats kept as pets. The sensitivity of ESBL *E. coli* to cefquinome was enhanced by garlic oil, indicating that garlic oil may be a potential antibiotic enhancer.

## 1. Introduction

As one of the most important and common Gram-negative bacteria (GNB) living in the gut of humans and animals, *Escherichia coli* (*E. coli*) can lead to severe diarrhea. *E. coli* can be transmitted between humans and animals, especially between pets and their owners [1,2,3,4]. With the occurrence of extended-spectrum β-Lactamases (ESBLs), bacterial resistance to β-lactams, especially cephalosporin, is becoming increasingly serious [5]. Research to explore antibiotic resistance mechanisms and methods to eliminate bacterial antimicrobial resistance has thus become high priority worldwide [6].

Several systematic reviews have described the complex mechanisms leading to antibiotic resistance, which can be mediated by plasmids, changes in target sites, modifications of antibiotic-degrading enzymes, cell adaptation, and efflux pumps, all of which have been linked to the inappropriate use of antibiotics [7,8]. Thus, the identification of antibiotic alternatives or synergistic approaches to reduce resistance is of great significance. There are studies which indicate that combinations of Chinese herb extracts and antibiotics show synergistic effects against *E. coli* via different mechanisms. A range of volatile oils from Cukangchai (*Mallotus philippensis* [*Lam.*] *Muell*. *Arg*.) inhibit the conjugal transfer of drug-resistant plasmids, which reduces the lateral transmission of drug-resistance [9]. Quercetin could cause MDR *E. coli* to regain susceptibility to tetracycline by increasing cell permeability and the intracellular drug concentration [10]. There are also studies that show that baicalin from Huangqin (*Scutellaria baicalensis Georgi*) inhibits the activity of *NDM-1* and decreases the expression of *fimB*, which is a major bacterial adhesion factor [11,12]. Resveratrol from Lilu (*Veratrum nigrum* L.) reduces the expression of the efflux pump protein AcrAB-TolC in *E. coli* to inhibit drug resistance [13].

Garlic oil is an extractive from garlic (*Allium sativum* L.), which was recorded in the *Ben Cao Gang Mu* (*Compendium of Materia Medica*, edited by Li Shizhen between 1522–1578, Mind Dynasty) to invigorate the spleen and stop diarrhea, has the functions of detoxification, detumescence, removing wind, and breaking a cold according to the theory of Traditional Chinese Medicine. Garlic is also a plant used as homology in medicine and food by Chinese people. As shown in previous studies, garlic oil has multiple biological activities, including the prevention and amelioration of cancer [14], improvement of the cardiovascular system [15], and anti-inflammatory effects [16]. In addition, some previous studies showed that garlic and garlic extracts were able to decrease the resistance of some bacteria to antibiotics, such as methicillin-resistant staphylococcus aureus [17] and ampicillin-resistant *E. coli* [18]. Moreover, garlic oil has been shown to regulate composition of gut flora [19]. Based on that, in this study, garlic oil was hypothesized to be able to reduce *E. coli* ESBL-producers.

Our previous study on canine-origin feces *E. coli*, isolated from multiple cities in Shaanxi Province, showed that there was multidrug-resistance of *E. coli* to cefquinome, and the resistance was eliminated by magnolol [20]. To further understand the prevalence of pet-origin drug-resistant *E. coli* in different species of companion animals (such as cats) in a wider area, the sampling range in this study was expanded to seven cities in different regions of China, and feline feces were collected as the sampling source.

All in all, this study aimed to detect the prevalence of ESBL feline-origin *E. coli* in seven cities in China to explore the elimination effect and possible mechanism of garlic oil in the resistance of ESBL *E. coli*, focusing on the membrane structure.

## 2. Results

### 2.1. Samples and E. coli Isolates

A total of 80 *E. coli* strains were isolated from 101 feline fecal samples obtained using anal swabs, for a rate of 79.21% (Figure 1). As shown in Appendix A 8 ESBL genes were detected. As shown in Figure 2C, 42 isolates (54.50%) were confirmed as ESBL *E. coli*.

In general, as shown in Figure 2A, the detection rate of *TEM* was the highest of tested genes at 100%, which is similar to our previous study [20]. The detection rate of *CTX-M-1* group genes was 36.25%, which was the second highest. The third highest detection rate was *CTX-M-9* group genes at 22.5%. The detection rates of *CTX-M-2* group genes, *OXA-1*, and *OXA-10* were similar at 2.5%, 2.5%, and 1.25%, respectively. *CTX-M-8*, *CTX-M-25*, *SHV*, and *OXA-2* were not detected in this study.

The geographical ESBL gene detection rates in seven cities were similar, between 13% to 23%. The rate was 20.00% in Tangshan, 19.52% in Shenyang, 16.43% in Zhengzhou, 15.00% in Lanzhou, 16.00% in Guangzhou, 12.86% in Kunming, and 12.86% in Shanghai.

### 2.2. Antibiotic Susceptibility Testing and ESBL Confirmatory Test

The nine tested antibiotics were all β-lactams, including cephalosporin (cefoxitin, ceftriaxone, ceftazidime, cefotaxime, cefixime and cefquinome), penicillin (amoxicillin, ampicillin), and carbapenem (meropenem). As shown in Figure 2B and Appendix A, the carbapenem class of antibiotics had the lowest resistance rate (17.50%). In contrast, the resistance rate of the detected penicillin class of antibiotics (93.75%) was highest. The resistance rate of cephalosporin was 71.25%. Furthermore, the resistance rates of detected 2nd, 3rd, and 4th generation cephalosporin antibiotics were 38.75% (cefoxitin), 76.25% (ceftriaxone, ceftazidime, cefotaxime, and cefixime), and 85% (cefquinome), respectively.

ESBL confirmatory tests (PCR method) were performed on isolates suspected to of ESBL *E. coli.* After PCR-positive products were sequenced and confirmed, as shown in Table 1, genes *CTX-M-1*, *15*, *230*, and *254* in *CTX-M-1* family were detected; *CTX-M-14*, *27* and *104*, in *CTX-M-9* family were detected; and genes *TEM-116* in *TEM-1* family were detected. As shown in Figure 2C, the detection rate of ESBL *E. coli* was 52.50% (42/80). As shown in Figure 2D, two isolates (4.76%) carried three ESBL genes simultaneously. Ten isolates (23.81%) carrying two ESBL genes were detected. As shown in Figure 2E, the ESBL rate in Shenyang was the highest in this study at 76.19% (16/21). All of the rates from Zhengzhou, Tangshan, and Guangzhou were more than 50%, at 57.14% (8/14), 66.67% (6/9), and 60% (3/5), respectively. The detection rates of ESBL *E. coli* were relatively low in Lanzhou, Kunming, and Shanghai, at 40% (4/10), 28.57% (2/7), and 21.43% (3/14), respectively. To analyze the genetic relationship of ESBL isolates geographically, the phylogenetic tree is mapped. As shown in Table 1, the main epidemic types of ESBL *E. coli* in China were *CTX-M-1*, *CTX-M-14*, and *TEM-116.* As shown in Figure 3, ESBL *E. coli* isolates in this study do not appear to be geographically related.

### 2.3. Synergistic Effect of Garlic Oil in Combination with Cefquinome

To evaluate the potential synergistic effect of garlic oil combined with cefquinome, checkerboard dilution assays were performed against 16 ESBL *E. coli* strains detected in this study. The FICIs of the combination against isolates are shown in Table 2. Notably, the FICI of ATCC^®^ 25922^TM^ was <0.5, indicating that there was synergistic effect between garlic oil and cefquinome. The rate of synergistic effect was 68.75%. The rate of additive effect was 31.25%. Furthermore, the use of cefquinome in combined treatment decreased 16- to 32-fold compared to monotherapy, suggesting that garlic oil eliminates the resistance of ESBL *E. coli* to cefquinome in the study.

### 2.4. Garlic Oil Enhanced Effects of Cefquinome on Killing ESBL E. coli

Based on the results of MIC assays, time-kill curves were performed to evaluate the bactericidal effect of cefquinome against ESBL *E. coli* treated with garlic oil. As shown in Figure 4A, compared with either the single cefquinome group or the single garlic oil group, at all concentrations tested, the combination of garlic oil and cefquinome exhibited an enhanced bactericidal effect against the three tested ESBL *E. coli* strains within 24 h. From zero to eight hours, the population of *E. coli* strains in the low concentration combination group were 10^2^- to 10^3^-fold compared to CEF group. At the 24 h point, the differences in the population between the GAR + CEF group (the low concentration group) and the CEF group reached 10^5^- to 10^6^-fold. Moreover, the bactericidal effect appeared to be dose dependent, as demonstrated by the phenomenon that the group with a level of 0.5 MIC garlic oil combined with 0.25 MIC cefquinome killed almost all bacteria within six to eight hours, which indicated that garlic oil may be a potential antibiotic activator. The above results suggest that garlic oil exerts an effective and rapid bactericidal effect on ESBL *E. coli*.

### 2.5. Garlic Oil Enhances the Ability of Cefquinome to Inhibit Growth of ESBL E. coli

Based on the MICs of three ESBL isolates, the growth curves were performed to analyze the effects of the combination of garlic oil and cefquinome on inhibiting growth of ESBL *E. coli*. As shown in Figure 4B, compared with either the CEF group or the GAR group, the CEF + GAR group showed better effects on inhibiting the growth of ESBL *E. coli* within seven hours, similar to the negative control. The above results suggest that the combination of garlic oil and cefquinome may be an ideal bacterial-inhibiting combination.

### 2.6. Garlic Oil Restores the Sensitivity of ESBL E. coli to Cefquinome

Based on the results of MICs determined in generations 0, 1, 2, 3, 6, 9, and 15, drug-resistance curves were prepared to evaluate the changes in drug resistance of ESBL *E. coli* treated with garlic oil over the course of 15 generations. As shown in Figure 5, the MICs in the GAR group kept decreasing within 15 generations and the downtrend of MICs in negative control group was not stable. Compared with the negative control group, the MICs of ESBL *E. coli* strains in the GAR group decreased more quickly in the first three generations (8- to 64-fold compared to the negative group). After 15 generations, the MICs in the GAR group were 12 times lower than the negative group. In generations 1, 2, 3, 6, and 9 of TS7 and SH2, the MICs were significantly lower than the negative group. In generation 1, 2, 3, and 15 of ZZ4, the MICs were significantly lower than in the negative group, as the standard that: * *p* < 0.05 (difference), ** *p* < 0.01 (significant difference), *** *p* < 0.001 (significant difference), **** *p* < 0.0001 (significant difference). Moreover, after two-way ANOVA tests, besides significant influence caused by garlic oil treatment or different generations, there was an interaction between two factors that could lead to significant changes in MICs, which indicated that the resistance of ESBL *E. coli* to cefquinome may be reduced with long-term garlic oil treatment. These results collectively suggested that garlic oil could reduce the resistance of ESBL *E. coli* to cefquinome.

### 2.7. Effects of Garlic Oil Combined with Cefquinome on Membrane Destruction

PI staining and NPN staining were used to detect the cell membrane integrity and permeability. When cell inner membrane integrity was damaged, the nucleic acid in cells could be stained by PI, and then PI would release fluorescence. As shown in Figure 6A, the fluorescence intensity (FI) in combination groups was significantly higher than the monotherapy groups, which indicated that garlic oil could enhance the destructive effects on inner membranes against ESBL *E. coli*. Furthermore, the high level of garlic oil (0.5 MIC) combined with the same level of cefquinome exhibited more effective destruction on inner membranes, which inferred that the destructive effects of the combination of garlic oil and cefquinome on the inner membrane were dose dependent.

On the other hand, NPN staining is a method to detect the permeability of cell outer membranes. When the permeability of cell outer membrane decreases, the NPN would combine with the hydrophobic parts of the phospholipid bilayer and release fluorescence. As shown in Figure 6B, the FI in combination groups was also significantly higher than the monotherapy groups, and simultaneously, the FI in the combination group with 0.5 MIC garlic oil was higher than the group with 0.25 MIC garlic oil, which suggested that garlic oil had the ability to reduce the permeability of the ESBL *E. coli* outer membrane.

Thus, results of PI and NPN staining indicated that garlic oil could enhanced the ability of cefquinome to destroy either the inner membrane or outer membrane of ESBL *E. coli*, which suggested that garlic oil may be an effective antibiotic enhancer.

### 2.8. Scanning Electron Microscope (SEM)

To verify garlic oil can enhance the ability of cefquinome to destroy the cell inner and out membrane of ESBL *E. coli*, scanning electron microscopy (SEM) was used to observe the morphological changes in one ESBL *E. coli* (ZZ4) after treatment with garlic oil. As shown in Figure 7, the surface of cells treated with cefquinome combined with garlic oil showed depression, shrinkage, and even collapse and lysis compared with monotreatment and the negative control treatment, indicating that the antibacterial effect of garlic oil may be related to the rapid destruction of cell surface structure.

## 3. Discussion

The emergence of ESBL *E. coli* has become a worldwide public health concern [5]. Numerous surveillance studies of ESBL bacteria in human and veterinary medicine have demonstrated that ESBL *E. coli* is associated with an increased risk of transmission and poses a significant threat to the sale of food products and public health [21,22]. To understand the current situation of ESBL *E. coli*, we examined the presence of 10 ESBL genes in 80 *E. coli* strains isolated from seven cities from different areas of China, including Shenyang (northeastern China), Lanzhou (northwestern China), Tangshan (northern China), Zhengzhou (middle of China), Shanghai (eastern China), Kunming (southwestern China), and Guangzhou (southern China). To the best of our knowledge, this is a very rare study on the prevalence of ESBLs on such a large geographical scale (Figure 2E).

The results of this study showed that the rate of ESBL *E. coli* was similar with previous studies in China [4,23]. However, the rates in different cities were totally different. For example, the ESBL rate in Shenyang was 76.19%, but the rate in Shanghai was 21.43%, which is nearly three times the difference between them. However, the rates in Guangzhou and Tangshan were similar at 60%. This phenomenon was also observed in our previous study [20]. In that study, we tried to raise several hypotheses to explain it: firstly, after communicating with local veterinarians, the hypothesis that the variations may be related to local and individual medication habits was verified. For instance, veterinarians in Shanghai preferred aminoglycosides such as amikacin and gentamycin to β-lactams such as ampicillin and ceftriaxone sodium during their daily practice. Thus, the rate of ESBL *E. coli* was lowest in Shanghai.

Secondly, we tested the hypothesis of geographical reason. The results of this study were compared with those of previous studies in China. We were surprised to find regional differences in the detection rate of ESBLs and the types of ESBL *E. coli* compared with previous studies [24,25]. However, a detailed phylogenetic comparison found that such geographical differences do not seem to be verified in this study. Nevertheless, we found that there were major epidemics in different cities, for example, the major epidemics of ESBL *E. coli* in Shenyang was *CTX-M-1*, while in Tangshan it was *CTX-M-15.* Combined with previous studies in a number of other countries, ESBL *E. coli* was found to be endemic to *CTX-M-1* in France, *OXA-48* in Switzerland, and *CTX-M-8* in Brazil [26,27,28]. Therefore, we revised our former hypothesis: the prevalence does depend on geography, and the type of epidemic is geographical.

Public policies may be another factor that could affect the prevalence of ESBL *E. coli*. In Europe, with guidelines on the prudent use of antimicrobial veterinary medicines published in 2015, the rates of ESBLs were relatively lower compared with China where appeals to reduce and replace antibiotics with other agents in veterinary medicine did not occur until 2020 [29,30,31,32]. The degree of social and economic development can also affect the prevalence of antibiotic resistance. It is clear that the ESBL rate in Shanghai, which is the most developed city in this study, was lowest. Furthermore, compared with previous studies in Africa and west Asia [33,34,35], the rates of ESBL genes in this study were lower. The impact of social and economic development can also be verified through comparisons with wealthier regions. The rates of ESBL-resistant genes in this study were slightly higher than the rates in west Europe and the United States [36,37,38,39].

In summary, four hypotheses may influence the prevalence of feline-origin ESBL *E. coli*: 1. local and individual medication habits; 2. geographic factors; 3. public policies; and 4. the degree of social and economic development. These hypotheses should be verified continuously in a follow-up study.

As previous studies found, garlic oil exhibited great antibacterial activity in vivo and vitro [40,41,42], and the activity was also verified in this study. However, unexpectedly, we found that garlic oil exhibited potent potentiation (4- to 16-fold) of the effectiveness of cefquinome against ESBL *E. coli* in a stable concentration (256~1024 μg/mL). To our knowledge, this study is the first to report the effect of the combination of garlic oil and cefquinome in inhibiting ESBL *E. coli*. Furthermore, to determine whether the antibacterial effect is related to garlic oil dose, we evaluated the bactericidal activity of high levels (0.5 MIC) of garlic oil combined with cefquinome. As indicated by time-kill curve analyses, the dose dependence was verified. The activity of garlic oil in inhibiting bacterial proliferation was also confirmed by the growth curve. Moreover, after continuous treatment with garlic oil, the MICs of ESBL *E. coli* to cefquinome decreased significantly. The significant decreases were proven to be related to two factors (generation and treatment of garlic oil) by two-way ANOVA test, and the interaction of two factors was confirmed. The above results clearly indicate that the combination of garlic oil and cefquinome has great antibacterial effects against ESBL *E. coli*, and more importantly, it is possible to reduce antibiotic resistance through continuous use of combinations of herbal extracts and antibiotics.

The cell membrane plays an important role in the metabolism and proliferation of bacteria [43,44], and previous studies found that bacteria was killed by cefquinome through damaging the cell membrane [45]. As a result of that, we hypothesized that there was a bactericidal effect by enhancing the cell membrane destruction effect of cefquinome when we combined garlic oil and cefquinome, which caused metabolic and replication disorders. To confirm our hypothesis, we tried to find some other previous studies on the antibacterial effect of plant extracts by altering membrane permeability. In the study on tetracycline and quercetin against multiple-drug-resistant *E. coli*, researchers found that the combination can kill MDR *E. coli* by altering cell membrane permeability [10], and moreover, they found that intracellular ATP levels decreased under the exposure of quercetin combined with tetracycline, which is consistent with our conjectures about the effect of drug combination on bacterial metabolism. Another two previous studies have found combinations of antibiotics and plant extracts can damage the integrity of cell membranes, thereby affecting their growth and proliferation and have a bactericidal effect [46,47], which fits our hypothesis about the effect of drug combination on bacterial replication.

Although it has been shown that the bacteriostatic effect would be achieved by damaging the integrity of cell membranes, the current studies are only apparent studies, and the mechanism of drug combinations damaging cell membranes has not yet been fully explored. Below are our hypotheses for possible patterns of drug combinations that damage cell membranes.

To study the possible mechanism of the antibacterial effect of garlic oil combined with cefquinome, we evaluated the cell membrane destruction activity of the combination through PI, NPN staining, and SEM, as conducted in a previous study [48]. After being treated with the garlic oil and cefquinome together, the permeability and integrity of both outer and inner membranes were damaged significantly and the destruction effects were proved to be garlic oil dose dependent, which preliminarily proved that the bactericidal effect might be related to the destruction of cell membranes, and the hypothesis was further proved through direct observation by SEM. As shown in Figure 7 and described in Figure 8, both the inner and outer membrane were complete and fluid in negative control and the plasmid DNA could not outflow. In cefquinome treatment, the outer membrane became smoother (shown in Figure 7), the permeability of the outer membrane decreased, and phospholipid gap enlarged (described in Figure 8), but the inner membrane was still complete, which we assumed was related to the hydrolytic action of ESBL enzymes in the periplasm. In garlic oil treatment, bean-shaped bulges that may be caused by the phospholipids affinity with garlic oil molecules occurred on the surface of the cell membrane, which contributed to enlargement of phospholipid gaps. In the combination treatment, the simultaneous action of garlic oil and cefquinome on the outer membrane caused significant damage to the outer membrane of the cell, resulting in a lot of cefquinome molecules entering the membrane gap, so the ESBLs could not hydrolyze cefquinome immediately. The cefquinome and garlic oil then interact with the inner membrane simultaneously, resulting in severe damage to the inner membrane, contributing to the destruction of ATP synthesis site and DNA outflow. As a result, metabolism and replication of bacteria were disrupted and the bacteria were killed. All hypotheses were based on the results of PI staining, NPN staining, and direct observation through SEM. However, the above hypotheses are still only based on the direct observation of a combination of garlic oil and cefquinome damaging cell membranes. Specific mechanisms should be further explored by transcriptome sequencing and other methods.

This study aimed to find a new method to fight ESBL *E. coli* infection in veterinary practice. Whether garlic oil can be used on cats was predicted by referring to several previous studies. First, whether garlic oil is poisonous to cats is the most important factor that should be taken into account. As described by other researchers, onion does show toxicity to cats because of dipropyl disulfide which is alike in structure to garlic oil at a dose of 5 g/kg [10], so we speculate that garlic oil may not be toxic to cats in certain doses. To verify our guess, we found another study which found that allicin (garlic oil in solid form) can improve the hemorheology of cats during anesthesia at a dose of 0.1 to 1 mg/kg, which is far below the toxic dose [49,50]. Combined with the results of antibacterial tests in vitro in this study, we speculated that doses of garlic oil combined with cefquinome would be far below the toxic dose. However, more clinical trials are needed to confirm whether the use of garlic oil on cats for bacterial infectious diseases is safe. Secondly, due to its unique odor and pungent taste, garlic oil still needs to find an effective method of flavor correction before it can be used in veterinary practice.

Although we have demonstrated the existence of ESBL *E. coli* in cats in different areas of China from feline fecal samples, to understand well the prevalence of feline origin ESBL *E. coli* better in China, the scale, scope, and time span of sampling can be expanded. The specific mechanism of drug combination on cell membrane destruction remains to be studied. Most importantly, the safe and effective dose to treat ESBL *E. coli* infection needs to be studied carefully.

This study has some limitations. First, the sample size in the different cities was relatively small; thus, larger samples will be needed to verify the four above-mentioned hypotheses. Second, despite the observed synergism of the combination of garlic oil and cefquinome, the mechanism of the synergy remains to be explored. Transcriptome sequencing has been used in drug-resistance reduction studies to identify changes in the expression of drug-resistance genes, data which can in turn be used in further studies to elucidate the effect of garlic oil on damaging cell membranes and the expression of ESBL genes. Furthermore, molecular docking and Western blotting can be used to evaluate the inhibitory effects of drugs on the activity and expression of ESBLs and cell membrane proteins.

In conclusion, our data indicate that ESBL *E. coli* strains are present in notable rates in cats kept as pets in China, which may raise public health concerns. Our study also demonstrated the elimination effects of garlic oil on cefquinome against ESBL *E. coli*. However, the mechanism of this synergistic activity remains to be elucidated in future studies. The discovery of garlic oil as a novel cefquinome adjuvant highlights the enormous antibacterial potential of compounds extracted from herbs.

## 4. Materials and Methods

### 4.1. Sample Collection

In this cross-sectional study, 103 feline-origin fecal samples were collected from 10 animal hospitals in 7 cities from different areas in China (Shenyang—northeast China, Tangshan—north China, Zhengzhou—middle of China, Lanzhou—northwest China, Shanghai—east China, Guangzhou—south China, and Kunming—southwest China) from September 2022 to February 2023, which were numbered by the first letter of the sampling city and the patient number.

### 4.2. Bacterial Isolation and Molecular Confirmation

All collected samples were enriched in trypticase soy broth for 10~12 h at 37 °C until reaching the logarithmic phase and then transferred onto MacConkey agar and then eosin-methylene blue agar and incubated aerobically for 16~18 h at 37 °C. One characteristic colony from each sample (red isolates on MacConkey agar an the black with metallic luster isolates) was selected and saved for the next steps. The *E. coli* isolates were subjected to Gram staining followed by primary identification. All media were purchased from Qingdao Hope Biotechnology Co., Qingdao, China. The *E. coli* strain ATCC^®^ 25922™ preserved in our laboratory was used as a control.

Single, pure isolates were enriched for a second time in Mueller–Hinton broth (MHB) for 24 h at 37 °C. Thereafter, 1 mL of bacterial culture was centrifuged at 14,000 rpm for 15 min. After decanting the supernatant, the pelleted cells were washed with sterile ultrapure water, and the centrifugation and wash steps were repeated twice. To extract genomic DNA, washed bacteria were boiled in sterile ultrapure water for 10 min. After centrifugation at 14,000 rpm for 15 min, the resulting supernatant was used as the DNA template for polymerase chain reaction (PCR) assays [4,20] using primers specific for 16S rDNA to identify the isolates, as described previously [51,52,53]. PCR assays were performed in a final volume of 20 μL, consisting of 10 μL of master mix (Dining, Beijing, China), 1 μL of each forward and reverse primer, 1 μL of DNA template, and 7 μL of nuclease-free water. PCR assays were performed in a thermocycler (Bioer TC-XP-G, Hangzhou, China) using the following program: initial denaturation at 94 °C for 5 min, followed by 30 cycles of denaturation at 94 °C for 30 s, annealing (Table 3) for 30 s, extension at 72 °C for 30 s, followed by a final extension at 72 °C for 7 min. The positive and negative controls were *E. coli* ATCC^®^ 25922™ and nuclease-free water, respectively. Electrophoresis was performed on a 1.5% agarose gel stained with DiRed Safe DNA DYE (Dining, China) to determine the size of PCR products compared to a 2000-bp DNA ladder. The gel was scanned using a UV-light transilluminator (72/BR04467, Bio-Rad, Hercules, CA, USA). Confirmed isolates were stored at −80 °C in MHB containing 35% glycerol until further analysis.

### 4.3. Antibiotic Susceptibility Testing

Broth-microdilution assays were performed to determine the antibiotic susceptibility and minimum inhibitory concentrations (MICs) for 9 β-lactam antibiotics and garlic oil (≥99%, Shanghai Macklin Biochemical Co., Ltd., Shanghai, China, Appendix A), including cefoxitin, ceftriaxone, ceftazidime, ceftriaxone, cefixime, cefquinome, meropenem, amoxicillin, and ampicillin (Shanghai Macklin Biochemical Co., Ltd., China) as recommended by the Clinical and Laboratory Standards Institute (CLSI 2023) and the European Committee on Antimicrobial Susceptibility Testing (EUCAST 2020) [55,56]. The above antibiotics are commonly used in veterinary clinical treatment of *E. coli* infection. Susceptibility to cefquinome was determined in reference to previous research [57]. All drugs were diluted 2-fold in MHB and mixed with an equal volume of bacterial suspension in a 96-well microtiter plate. Each test was repeated 3 times. *Escherichia coli* ATCC^®^ 25922™ was used as the quality-control strain.

### 4.4. ESBL Confirmatory Test

The PCR method was used for the ESBL confirmatory test. All primers were reflected in Table 3. PCR assays were performed in a final volume of 20 μL consisting of 10 μL of master mix (Dining, China), 1 μL of each forward and reverse primer, 1 μL of DNA template, and 7 μL of nuclease-free water. PCR assays were performed in a thermocycler (Bioer TC-XP-G, China) using the following program: initial denaturation at 94 °C for 5 min, followed by 30 cycles of denaturation at 94 °C for 30 s, annealing (Table 1) for 30 s, extension at 72 °C for 30 s, followed by a final extension at 72 °C for 7 min. The primers refer to some previous studies [54,58,59,60,61,62,63,64]. The negative control was nuclease-free water. Electrophoresis was performed on a 1.5% agarose gel stained with DiRed Safe DNA DYE (Dining, China) to determine the size of PCR products compared to a 2,000-bp DNA ladder. The gel was scanned using a UV-light transilluminator (72/BR04467, Bio-Rad, Hercules, CA, USA). PCR positive products were sent to Tsingke Biotechnology Co., Ltd., Beijing, China. for sequencing and compared with the NCBI gene bank to confirm whether they were positive for ESBLs.

### 4.5. Checkerboard Assay

The combined antibacterial effect of garlic oil and cefquinome was assessed using a checkerboard assay, as previously described [65]. Briefly, both garlic oil and cefquinome were diluted to prepare 7 gradient concentrations ranging from 1/16 MIC to 2 MIC. Each longitudinal column of tubes contained the same concentration of drug A, and each horizontal row of tubes contained the same concentration of drug B. Each tube was inoculated with bacterial suspension to a final density of approximately 1 ×  10^6^ CFU/mL. Single-drug control tubes and blank control tubes were also prepared, and *E. coli* ATCC^®^ 25922™ was used as a sensitivity control strain. Six ESBL isolates were used as experimental bacteria. All tubes were incubated at 37 °C for 16 h under aerobic conditions. The experiment was repeated in triplicate. The fractional inhibitory concentration index (FICI) was calculated according to the following formula (Table 4):

FICI  =  MIC of garlic oil in combination/MIC of garlic oil alone + MIC of cefquinome in combination/MIC of cefquinome alone.

In this study, synergy and partial synergy were defined as a synergistic relationship, whereas additive, indifferent, and antagonistic results were regarded as a non-synergistic relationship [66].

### 4.6. Time-Kill Curves

Time-kill assays were used to evaluate the antibacterial effects of the combination of garlic oil and cefquinome against ESBL *E. coli* by measuring the reduction in the calculated population in CFU/mL within 24 h. Garlic oil and cefquinome were incubated with an equal volume of *E. coli* culture at different levels of garlic oil and cefquinome [67]. As a control, MHB was added instead of garlic oil or cefquinome. All samples were cultivated at 37 °C. After 0, 2, 4, 6, 8, and 24 h of incubation, 100-μL samples were removed. After 10 rounds of centrifugation and resuspending to wash off residual medicine, proper dilutions were performed (Table 5), and 100 μL of each sample was spread onto Mueller–Hinton agar for colony counting. Each assay was repeated in triplicate.

### 4.7. Growth Curves

The growth curve was used to evaluate the growth inhibition effect of the combination on ESBL *E. coli* from the 7 h point until the logarithmic phase. Garlic oil and cefquinome were incubated with an equal volume of *E. coli* culture at 0.25 MIC of garlic oil and cefquinome. As a control, MHB was added instead of garlic oil or cefquinome. The starting concentration of bacterial culture was 1 × 10^6^ CFU/mL. All samples were cultivated at 37 °C. Each hour during incubation, 100-μL samples were removed to measure the absorbance at OD_600_. Considering the emulsification of garlic oil, 5 tubes without isolates were also incubated. We plotted the curves with the absorbance changes at OD_600_ over time. Each assay was repeated in triplicate.

### 4.8. Drug-Resistance Curves

Drug-resistance curves were used to evaluate the effects of garlic oil in reducing the resistance of ESBL *E. coli* to cefquinome by determining the MIC after garlic oil treatment within 15 generations. Garlic oil (0.25 MIC) was incubated with an equal volume of each *E. coli* culture in MHB at 37 °C for 16 h. An inoculating loop of each MHB culture was then streaked onto Mueller–Hinton agar and incubated at 37 °C for 16 h. After 0, 1, 2, 3, 6, 9, and 15 generations, a single, pure colony of each isolate was removed and placed in MHB and incubated at 37 °C for 16 h, after which the MIC was determined. Each assay was repeated in triplicate.

### 4.9. PI Staining and NPN Staining

PI staining and NPN staining were used to evaluate the destructive effects of garlic oil combined with cefquinome on the inner and outer membrane of *E. coli* [48]. A fluorescence intensity of 10 nM propidium iodide (PI)-labeled *E. coli* incubated in MHB with garlic oil monotherapy, cefquinome monotherapy, or the combination of garlic oil and cefquinome was measured with the excitation wavelength of 535 nm and emission wavelength of 615 nm. The concentration of all above drugs was 0.25 MIC. As a control, MHB was added instead of garlic oil or cefquinome. The same operations were used in NPN (N-Phenyl-1-naphthylamine) staining, but the fluorescence intensity was measured with the excitation wavelength of 350 nm and emission wavelength of 420 nm.

### 4.10. Scanning Electron Microscope (SEM)

Isolates were incubated in MHB with 0.25 MIC garlic oil monotherapy, 0.25 MIC cefquinome monotherapy, or in the combination of 0.25 MIC garlic oil and 0.25 MIC cefquinome for 10 h. As a control, MHB was added instead of garlic oil or cefquinome. After 10 h, the bacteria were collected and washed by PBS. The bacteria were then fixed with 4% glutaraldehyde for 3 h. After that, the bacteria were dehydrated by using graded ethanol. Before beginning scanning electron microscopy, carbon dioxide critical point drying and a gold spraying operation were performed on the bacteria.

### 4.11. Statistical Analysis

Data are expressed as the mean ± standard deviation. The statistical significance of differences was determined using a 1-way ANOVA test in PI and NPN staining and a 2-way ANOVA test in drug-resistance curves with SPSS 27.0 software. For all comparisons, *p* < 0.01 and *p* < 0.05 were considered indicative of statistical significance. All figures were made by GraphPad Prism 8.0.1. The phylogenetic tree was drawn by Geneious Prime. The maps were downloaded from Standard Map Service accessed on 1 March 2023 (http://bzdt.ch.mnr.gov.cn/) and edited by Photoshop 2021.

## Figures and Tables

**Figure 1 ijms-24-09627-f001:**
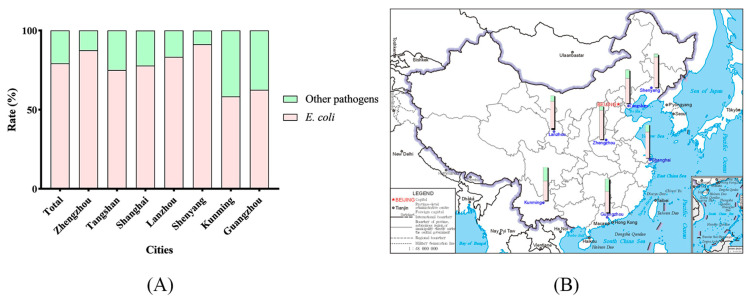
Collection and isolation of feline-origin *Escherichia coli* in seven cities in China. (**A**) Rate of *E. coli* isolates; (**B**) geographic rate of *E. coli* isolates.

**Figure 2 ijms-24-09627-f002:**
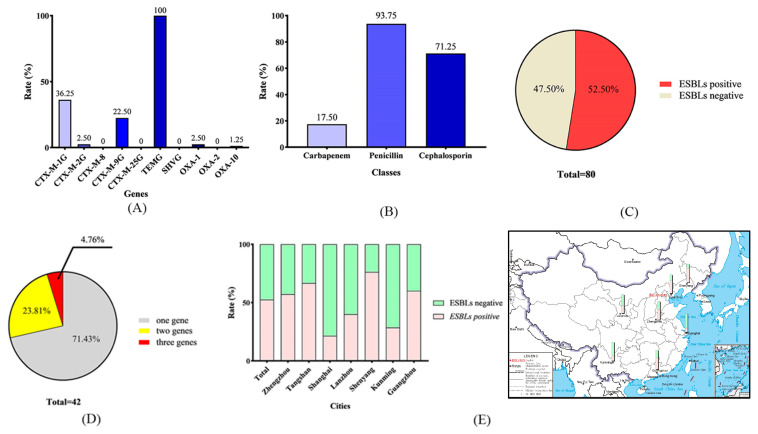
Results of ESBL detection of isolated *E. coli*. (**A**) Detection rates of 10 ESBL-resistant genes; (**B**) resistance of isolates to three kinds of β-lactam antibacterial drugs; (**C**) detection rate of ESBL *E. coli* in isolates; (**D**) rate of isolates with multiple ESBL genes; (**E**) geographic rate of ESBL *E. coli* isolates.

**Figure 3 ijms-24-09627-f003:**
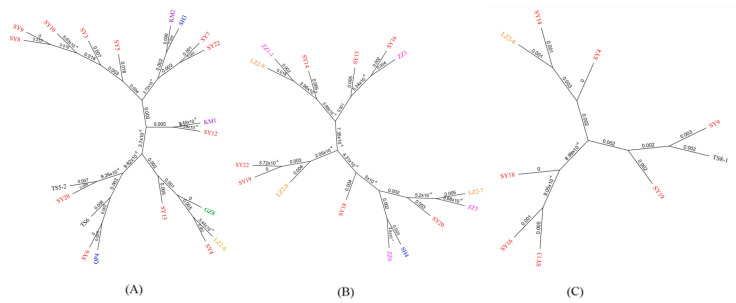
Phylogenetic tree of isolates carrying three groups of ESBL genes. (**A**) Genes in CTX-M-1 group; (**B**) genes in CTX-M-9 group; (**C**) genes in TEM group.

**Figure 4 ijms-24-09627-f004:**
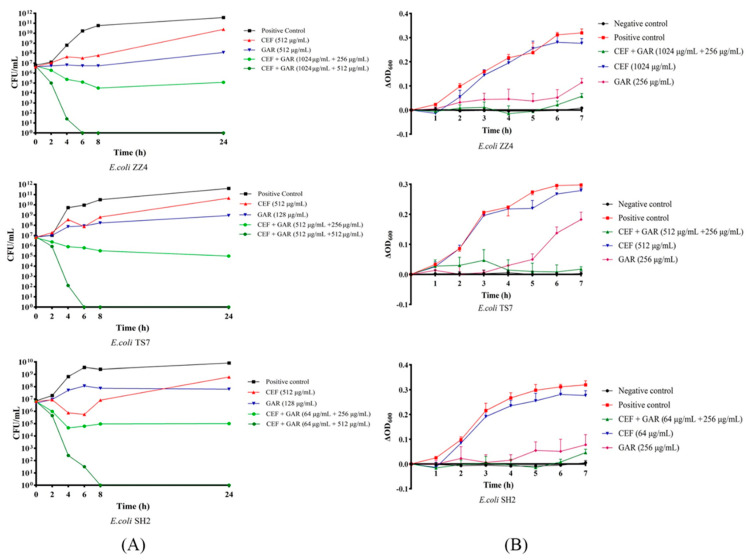
(**A**) Time-kill curves of garlic oil (GAR) and cefquinome (CEF) in different combinations or levels against ESBL *E. coli*. (**B**) Growth curves of ESBL *E. coli* treated with garlic oil (GAR) and cefquinome (CEF) in different combinations or levels.

**Figure 5 ijms-24-09627-f005:**
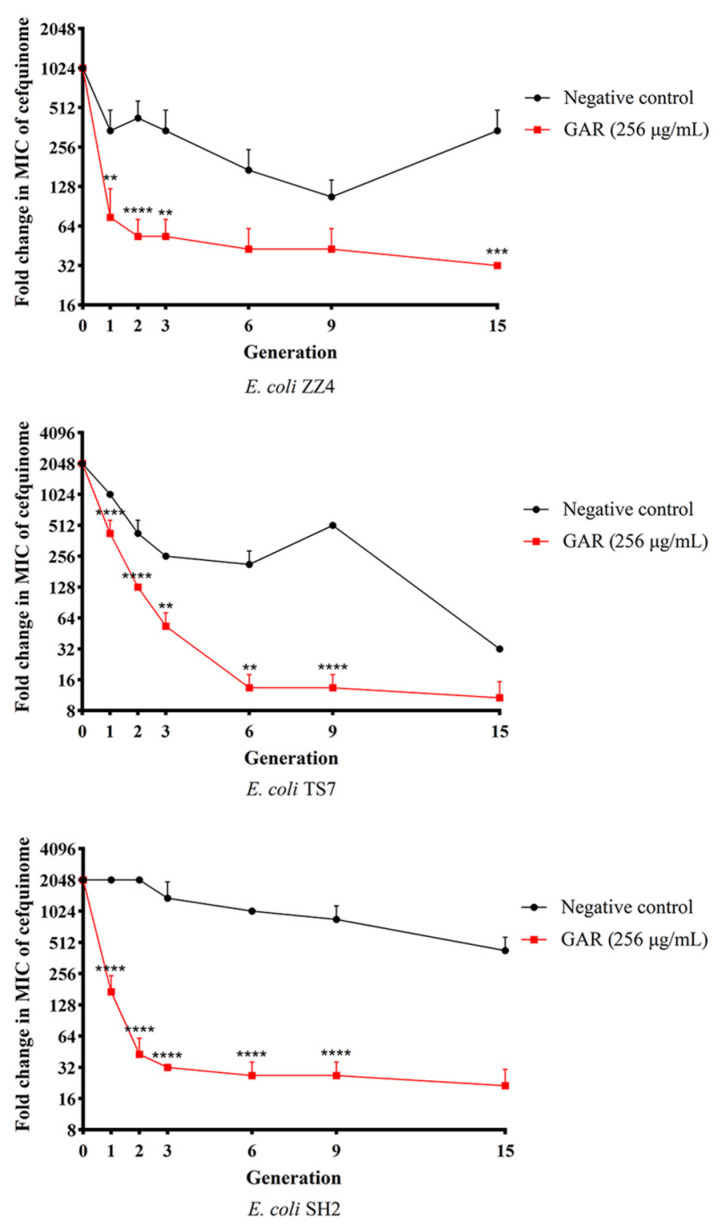
Effect of garlic oil (GAR) on changing the sensitivity of ESBL E. coli to cefquinome. All data are expressed as mean ± SD determined from three independent experiments performed in triplicate and significance was determined by a two-way ANOVA test. ** *p* < 0.01; *** *p* < 0.001; **** *p* < 0.0001.

**Figure 6 ijms-24-09627-f006:**
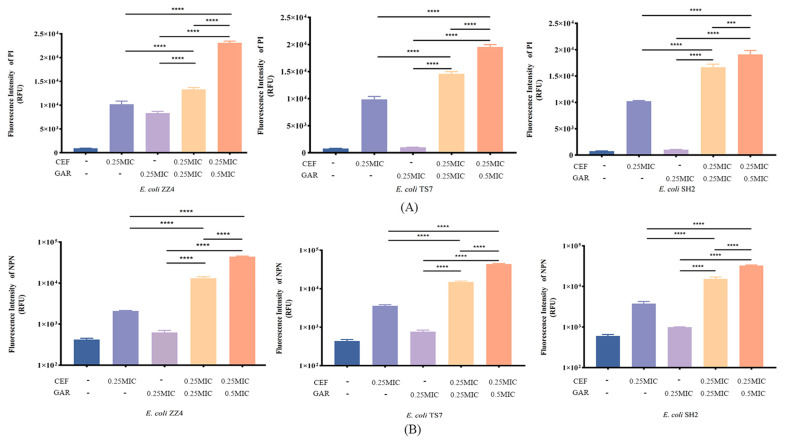
(**A**) Fluorescence intensity of PI staining of ESBL *E. coli* treated with garlic oil (GAR) and cefquinome (CEF) in different combinations or levels. (**B**) Fluorescence intensity of PI staining of ESBL *E. coli* treated with garlic oil (GAR) and cefquinome (CEF) in different combinations or levels. All data are expressed as mean ± SD determined from three independent experiments performed in triplicate and significance was determined by one-way ANOVA test. *** *p* < 0.001; **** *p* < 0.0001.

**Figure 7 ijms-24-09627-f007:**
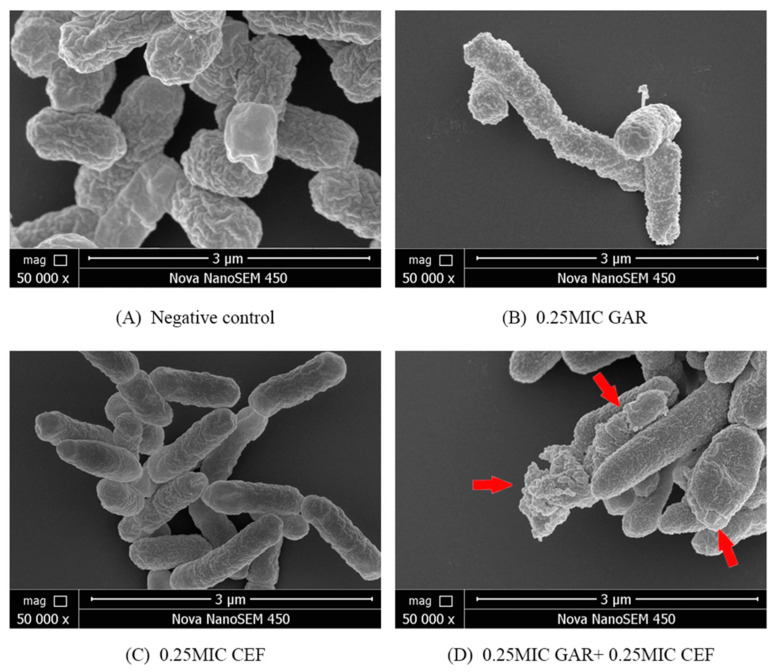
Morphological changes of ESBL *E. coli* (ZZ4) after different treatments. (**A**) Negative control; (**B**) 0.25MIC garlic oil (GAR); (**C**) 0.25MIC cefquinome (CEF); (**D**) 0.25MIC garlic oil (GAR) + 0.25MIC cefquinome (CEF).

**Figure 8 ijms-24-09627-f008:**
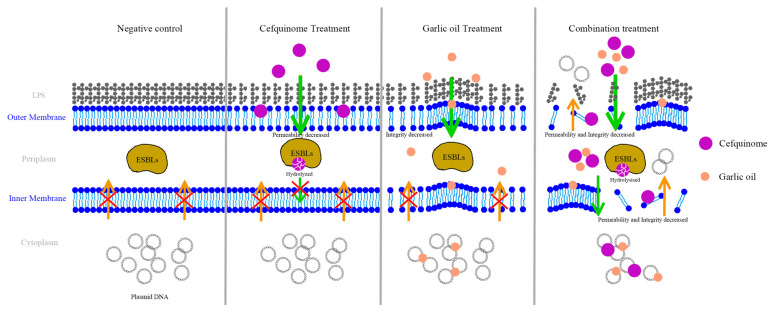
The hypothesized mechanism picture of inhibition effect of garlic oil combined with cefquinome against ESBL *E. coli* by destroying cell membranes.

**Table 1 ijms-24-09627-t001:** ESBL types of isolates.

ESBL Genes Group	Isolates	Type	ESBL Genes Group	Isolates	Type
*CTX-M-1*	GZ8	*CTX-M-1*	*CTX-M-9*	LZ2-7	*CTX-M-27*
	KM1	*CTX-M-1*		LZ2-8	*CTX-M-14*
	KM2	*CTX-M-1*		LZ2-9	*CTX-M-104*
	LZ2-6	*CTX-M-15*		SH2	*CTX-M-14*
	QP4	*CTX-M-1*		SY13	*CTX-M-14*
	SH3	*CTX-M-254*		SY14	*CTX-M-14*
	SY10	*CTX-M-230*		SY16	*CTX-M-14*
	SY12	*CTX-M-1*		SY18	*CTX-M-14*
	SY13	*CTX-M-1*		SY19	*CTX-M-14*
	SY20	*CTX-M-1*		SY20	*CTX-M-27*
	SY22	*CTX-M-1*		SY22	*CTX-M-14*
	SY3	*CTX-M-230*		ZZ1-1	*CTX-M-14*
	SY4	*CTX-M-15*		ZZ3	*CTX-M-14*
	SY5	*CTX-M-1*		ZZ5	*CTX-M-27*
	SY6	*CTX-M-1*		ZZ6	*CTX-M-14*
	SY7	*CTX-M-1*	*TEM*	LZ2-6	*TEM-116*
	SY8	*CTX-M-1*		SY13	*TEM-116*
	SY9	*CTX-M-230*		SY14	*TEM-116*
	TS5-2	*CTX-M-15*		SY16	*TEM-116*
	TS6	*CTX-M-1*		SY18	*TEM-116*
	TS7	*CTX-M-1*		SY19	*TEM-116*
	TS8-2	*CTX-M-15*		SY4	*TEM-116*
	ZZ4	*CTX-M-1*		SY9	*TEM-116*
	ZZ7-1	*CTX-M-1*		TS8-1	*TEM-116*
	ZZ8	*CTX-M-1*			
	ZZ9	*CTX-M-1*			

**Table 2 ijms-24-09627-t002:** FICI results of MIC (garlic oil) and FICI (garlic oil × cefquinome) for seven *E. coli* isolates.

Strain	MIC (Alone)/μg/mL		MIC (Combined)/μg/mL		FICI	Outcome
	Garlic Oil	Cefquinome	Garlic Oil	Cefquinome		
ATCC^®^ 25922^TM^	512	0.5	128	0.125	0.5	Synergy
ZZ4	1024	2048	128	128	0.1875	Synergy
ZZ7-1	512	256	32	128	0.5625	Additive effect
ZZ8	512	256	256	64	0.75	Additive effect
ZZ9	512	2048	256	256	0.625	Additive effect
TS5-2	512	256	64	32	0.25	Synergy
TS6	512	1024	64	256	0.375	Synergy
TS7	1024	2048	128	128	0.1875	Synergy
GZ8	512	1024	64	256	0.375	Synergy
SY4	128	512	32	32	0.3125	Synergy
SY9	1024	8	512	2	0.75	Additive effect
SY18	1024	8	256	2	0.5	Synergy
SY19	512	16	128	2	0.375	Synergy
SH2	1024	256	32	64	0.28125	Synergy
LZ2-3	512	1024	64	128	0.25	Synergy
GZ8	1024	2048	256	512	0.5	Synergy
KM1	2048	256	256	128	0.625	Additive effect

**Table 3 ijms-24-09627-t003:** Sequences of *E. coli* 16s rDNA and ESBL gene primers used in PCR assays.

Gene	Sequence of Primer (5′~3′)	Size of Product/bp	Tm/°C	Reference
16s rDNA	F:AGAGTTTGATCCTGGCTCAG	306	55.0	[19,20]
	R: CTTGTGCGGGCCCCCGTCAATTC			
*CTX-M Family*	F-ATGTGCAGYACCAGTAARGTKATGGC	592	55.0	[52]
	R-TGGGTRAARTARGTSACCAGAAYSAGCGG			
*CTX-M-1 Group*	F-ACCGCGATATCGTTGGT	550	55.0	[54]
	R-CGCTTTGCGATGTGCAG			
*CTX-M-2 Group*	F-ATGATGACTCAGAGCATTCG	856	55.0	[53]
	R-TCAGAAACCGTGGGTTACGA			
*CTX-M-8*	F-GTGACAAAGAGAGTGCAACGG	666	52.0	[55]
	R-ATGATTCTCGCCGCTGAAGCC			
*CTX-M-9 Group*	F-GCACGATGACATTCGGG	857	52.0	[56]
	R-AACCCACGATGTGGGTAGC			
*TEM Group*	F-ATGAGTATTCAACATTTCCG	858	55.0	[54]
	R-CCAATGCTTAATCAGTGAGG			
*SHV Group*	F-ATGAGTATTCAACATTTTCG	841	55.0	[53]
	R-TTACCAATGCTTAATCAGTG			
*OXA-1*	F-ATGCGTTATATTCGCCTGTG	820	55.0	[52,57]
	R-TTAGCGTTGCCAGTGCTCGA			
*OXA-2*	F-ATGAAAAACACAATACATATCAACTTCGC	601	55.0	[52,57]
	R-GTGTGTTTAGAATGGTGATCGCATT			
*OXA-10*	F-ACGATAGTTGTGGCAGACGAAC	277	55.0	[54]
	R-ATYCTGTTTGGCGTATCRATATTC			

**Table 4 ijms-24-09627-t004:** FICI values and criteria definitions.

FICI	Meaning
FICI ≤ 0.5	Synergistic effect
0.5 < FICI ≤ 0.75	Partial synergistic effect
0.75 < FICI ≤ 1	Additive effect
1 < FICI ≤ 4	Indifferent effect
FICI > 4	Antagonism

**Table 5 ijms-24-09627-t005:** Bacterial solution dilution ratio in time-kill curves.

Group	Incubation Time/h	Isolates	Dilution Ratio
Positive control	0	TS7, ZZ4 and SH2	10^5^
	2	TS7, ZZ4 and SH2	10^5^
	4	TS7, ZZ4 and SH2	10^7^
	6	TS7, ZZ4 and SH2	10^9^
	8	TS7, ZZ4 and SH2	10^9^
	24	TS7, ZZ4 and SH2	10^9^
CEF	0	TS7, ZZ4 and SH2	10^5^
	2	TS7, ZZ4 and SH2	10^5^
	4	TS7 and ZZ4	10^6^
		SH2	10^4^
	6	TS7 and ZZ4	10^6^
		SH2	10^4^
	8	TS7 and ZZ4	10^9^
		SH2	10^7^
	24	TS7 and ZZ4	10^6^
		SH2	10^5^
GAR	0	TS7, ZZ4 and SH2	10^5^
	2	TS7, ZZ4 and SH2	10^5^
	4	TS7, ZZ4 and SH2	10^5^
	6	TS7, ZZ4 and SH2	10^5^
	8	TS7, ZZ4 and SH2	10^5^
	24	TS7, ZZ4 and SH2	10^6^
COM (low)	0	TS7, ZZ4 and SH2	10^5^
	2	TS7 and ZZ4	10^5^
		SH2	10^4^
	4	TS7 and ZZ4	10^4^
		SH2	10^3^
	6	TS7 and SH2	10^3^
		ZZ4	10^4^
	8	TS7 and SH2	10^3^
		ZZ4	10^4^
	24	TS7 and SH2	10^3^
		ZZ4	10^4^
COM (high)	0	TS7, ZZ4 and SH2	10^6^
	2	TS7, ZZ4 and SH2	10^4^
	4	TS7, ZZ4 and SH2	10^1^
	6	TS7, ZZ4 and SH2	10^0^
	8	TS7, ZZ4 and SH2	10^0^
	24	TS7, ZZ4 and SH2	10^0^

## Data Availability

Data is contained within the article or Appendix A. The data presented in this study are available in [Detection of Antibiotic Resistance in Feline-Origin ESBL *Escherichia coli* from Different Areas of China and the Resistance Elimination of Garlic Oil to Cefquinome on ESBL *E. coli*].

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
