# Peer review of "Detection of Antibiotic Resistance in Feline-Origin ESBL Escherichia coli from Different Areas of China and the Resistance Elimination of Garlic Oil to Cefquinome on ESBL E. coli"

_ijms, 2023, doi:10.3390/ijms24119627_

Round 1
Reviewer 1 Report
The topic of the manuscript is quite interesting and probably the conducted research is of application importance. However, the manuscript requires a few clarifications and corrections.
Please verify the manuscript for English, e.g. line67 (should be Moreover), line 69-E.coli ESBL-producers etc.
Lines 35-37: a sentence is too long, causing it to lose its meaning.
Figure S1/B: Error in Ampicillin name
The very high percentage of ESBL isolates is quite surprising, especially since the authors used media without cefotaxime supplementation. Were one characteristic colony selected from each sample for testing?
Line 120: Please include the CLSI standard in the reference list
2.5. Checkerbord assay: a tabelaric scheme would be very useful for the description, which will illustrate the analysis much better.
The beginning of the results was described quite vaguely - the authors should indicate at the beginning how many ESBL strains they isolated from the tested samples. The authors also indicated that for all strains they showed the presence of the TEM gene, but Table 3 does not confirm this
Why was the presence of cephalosporinases (AmpC) not also determined?
The authors presented the probable mechanism of the antibacterial effect of the substances used in the study (Garlic oil and cefquinome), therefore the conclusions should clearly indicate the limitations of the analyzes and interpretation of the results obtained
Please see comments
Author Response
Cover letter of revisions
Thank you for your patience and valuable suggestions. The following is the content of the revision. All revisions were highlighted in blue.
- As your advice, some inappropriate English expressions have been corrected. e.g. line 67, line 69 and etc.
- In line 36 – 39, the long sentence was re-written and was split into two setences.
- The Ampicillin name was corrected in Figure S1/B.
- The reason of the very high percentage of ESBL isolates was explained in the discussion section and was highlighted in blue around line 382 – 387. The selection of isolates was described in line 91 – 93.
- The CLSI standard and EUCAST standard were added in the reference list.
- The table of FICI standard criteria was listed in table 3, line 159.
- The number and rate of ESBLs E. coli were described clearly in line 223.
- For your questions about TEM detection, the following explanations are given. TEM gene was detected in all strains. However, there are several genotypes in TEM gene, and only a small part of genotypes belong to ESBLs genes. After sequencing the TEM amplification products of all strains, it was found that only 9 strains as shown in Table 4 carried TEM genes belonging to ESBLs genes.
- For your questions about AmpC, here’s the reasons. AmpC doesn’t belong to ESBLs genes, so we didn’t include it in the test.
- The limitations and conclusions were added in line 502 – 517.

Reviewer 2 Report
The manuscript aimed to detect the prevalence of feline-origin ESBLs E. coli in China as well as the elimination effect of garlic oil to cefquinome on ESBLs E. coli. The article is well written, based on scientific language, and good English level. Statistical analysis and illustrating data with tables were carefully added in this investigation.
However, this article has some aspects that must be improved. I recommend you include the following articles in the manuscript, https://www.ncbi.nlm.nih.gov/pmc/articles/PMC7874003/, and https://www.ncbi.nlm.nih.gov/pmc/articles/PMC8001562/ in the discussion section (around line 383). The authors compared the results regarding the prevalence of ESBL-producing E. coli with other cities in China, but it will be interesting to also mention differences with other countries and discuss them. Moreover, a subsection regarding sequencing must be included in the methods section (conditions, place,…). Please, check the references (bacteria names must be in italic style).
1 – Which do authors consider the novelty of this work, compared to previous data?
2 - I saw that the authors did not test colistin resistance (lines 117-118). Why?
Minor corrections are required as indicated down.
Line 36 – replace “diarrhea. Escherichia coli …”
Line 72, 73, 82,… – pay attention to spaces before ().
Line 120-121 – Include the abbreviations, year, and reference.
Line 219 – Include the prevalence of ESBL-producing isolates.
Line 250 - Replace to “As shown in Figure 2(D), 2 isolates (4.76%) carried 3 ESBLs genes simultaneously.”
Line 412 – Check if it is correct “16- to 32-fold”
References 9, 18, 19… - Don’t forget that bacteria names must be in italic style.
Minor comments. Please check the previous section.
Author Response
Cover letter of revisions
Thank you for your patience and valuable suggestions. The following is the content of the revision. All revisions were highlighted in blue.
- As your suggestions, the mentioned articles were referred in line 408 – 409.
- The differences of the prevalence of ESBL-producing E. coli between China and other countries were mentioned in line 391 – 409, including countries in west Europe, Asia, Africa and America.
- The work of sequencing was finished by Tsingke Biotechnology Co., Ltd. And relevant description was added in line 141 – 143.
- All bacteria names in reference were checked and changed into italic style.
- The novelty of this work was described in line 375 – 376.
- For your questions about the test of colistin, here’s the reason. As mentioned in title, this study aimed to detect the prevalence of ESBLs E. coli in China. However, colistin doesn’t belong to beta-lactam antibiotics, so we didn’t conduct the test.
- Line 36 –The expression was optimized.
- Line 72, 73, 82,… – Spaces were added before ().
- Line 120-121 – Include the abbreviations, year, and reference.
- Line 219 – The prevalence of ESBL-producing isolates was added in line 223.
- Line 250 – The expression was replaced to “As shown in Figure 2(D), 2 isolates (4.76%) carried 3 ESBLs genes simultaneously.”
- Line 412 – The data was verified and changed into “4- to 16- fold”
